# Recent Advances in Cancer Therapeutic Copper-Based Nanomaterials for Antitumor Therapy

**DOI:** 10.3390/molecules28052303

**Published:** 2023-03-01

**Authors:** Reyida Aishajiang, Zhongshan Liu, Tiejun Wang, Liang Zhou, Duo Yu

**Affiliations:** 1Department of Radiotherapy, The Second Affiliated Hospital of Jilin University, Changchun 130062, China; 2State Key Laboratory of Rare Earth Resource Utilization, Changchun Institute of Applied Chemistry, Chinese Academy of Sciences, Changchun 130022, China

**Keywords:** cancer cell, antitumor therapy, reactive oxygen species, copper-based nanomaterials, copper-associated cell death

## Abstract

Copper serves as a vital microelement which is widely present in the biosystem, functioning as multi-enzyme active site, including oxidative stress, lipid oxidation and energy metabolism, where oxidation and reduction characteristics are both beneficial and lethal to cells. Since tumor tissue has a higher demand for copper and is more susceptible to copper homeostasis, copper may modulate cancer cell survival through reactive oxygen species (ROS) excessive accumulation, proteasome inhibition and anti-angiogenesis. Therefore, intracellular copper has attracted great interest that multifunctional copper-based nanomaterials can be exploited in cancer diagnostics and antitumor therapy. Therefore, this review explains the potential mechanisms of copper-associated cell death and investigates the effectiveness of multifunctional copper-based biomaterials in the field of antitumor therapy.

## 1. Introduction

Copper (Cu) is a common metallic element and also a transition element characterized by redox activity. Cuprous (Cu^+^) can be reduced to oxidized cupric in regular chemical reaction under physical conditions [1]. Copper ions participate in several metabolic activities as cofactors or structural components, supplying or receiving electrons to regulate multiple physiological processes including energy metabolism, mitochondrial respiration and antioxidants [2]. The presence of copper ions keeps a balance that, if disturbed, could generate oxidative stress and abnormal autophagy [3,4]. Tumor tissue and blood copper levels have been demonstrated to be much higher in individuals with various malignancies, and abnormal copper involvement in patients with Wilson’s disease may promote the malignant transformation of hepatocytes [5]. Therefore, copper homeostasis plays a vital role in tumor cell survival and development.

Copper is commonly known to be essential for embryogenesis and cell proliferation. Cancer patients and tumors have elevated levels of copper in comparison with healthy ones, and copper concentration correlates with tumorigenesis, angiogenesis, tumor metastasis and recurrence in various human cancers. Thiele and colleagues discovered that the copper transporter-1 (CTR1) inhibited tumor cell proliferation through copper depletion by activating the mitogen-activated protein kinases (MAPK) pathway, which could control proliferation and division broadly, and that the chelator tetrathiomolybdate (TTM) inhibited tumor cell proliferation by significantly hindering its downstream kinase mitogen-activated protein kinase kinases 1 (MEK1) signaling [6,7]. As the demand for copper grows progressively throughout aberrant proliferation in cancers, copper depletion therapy may provide effective anticancer and anti-metastasis activity. Cui et al. proved that mitochondrial-targeted copper-depleting nanoparticle inhibited tumor growth and substantially improved survival of triple-negative breast cancer [8]. In addition, supplying copper could exert anticancer effects by copper cytotoxicity. Tsvetkov has recently proposed for the first time a copper-dependent form of cell death ‘cuproptosis’, in which copper homeostasis depends on the tricarboxylic acid cycle (TCA) during mitochondrial respiration, causing protein toxic stress and cell death [9]. For mitochondrial respiratory-dominated tumor cells, enhancing anti-tumor therapy by increasing the copper death pathway is also a promising anticancer approach.

Copper-based nanoparticles have inspired great interest in the area of antitumor therapy in recent decades, and their unique physicochemical features and remarkable biocompatibility make them suitable for biomedical applications, particularly tumor imaging and antitumor treatment. Copper-based nanoparticles have strong near-infrared absorption and significant photothermal capabilities, and they have been extensively employed in photothermal therapy and cancer photoimaging. In addition, copper-based nanomaterials provide a large specific surface area that may be applied to load multiple antitumor drugs. Moreover, they also generate a large number of reactive oxygen species (ROS) when exposed to light, which could be adopted for photodynamic therapy. As a common metal, copper provides considerable advantages in cancer treatment and transformation potential by exploiting its unique bioactivity, convenient synthesis procedures, low response conditions, and high yields. Therefore, the biological applications and possibilities of copper-related cell death and copper-based nanomaterials are the focus of this review.

## 2. Antitumor Bioactivity of Copper Nanomaterials

The essence of oxidative stress (OS) is the imbalance state of oxidation–antioxidation system, manifested by enhanced ROS concentration instantly or chronically [10]. Most cancer cells exhibited increased aerobic glycolysis and high levels of oxidative stress [11]. However, such amounts were less toxic than they would be in normal cells. To be specifical, persistently low levels of OS may promote cellular proliferation and tumor migration, whereas high concentrations of OS might trigger cancer cell death [12]. As a cancer-theragnostic transition metal, increasing Cu ions in cancer tissue makes an antitumor impact that mainly involves OS by triggering the Fenton reaction, which can produce ROS. Cu^2+^ is vulnerable to the reduction to Cu^+^, allowing Cu to drive the Fenton reaction and produce hydroxyl radicals (·OH). Furthermore, compared with other metals (iron, chromium, cobalt and nickel), the Cu-based Fenton reaction can react in wider pH range with higher reaction rate (k = 1.0 × 10^4^ M^−1^ s^−1^) [13]. The Cu-based catalytic reaction can be given in Equations (1)–(3).
(1)$${\mathrm{Cu}}^{+}{+H}_{2}O_{2}\to {\mathrm{Cu}}^{2+}{+\mathrm{OH}}^{-}+\mathrm{OH}$$
(2)$${\mathrm{Cu}}^{2+}{+H}_{2}O_{2}\to {\mathrm{Cu}}^{+}{+\mathrm{OH}}^{-}{+\mathrm{OH}}_{2}{}^{-}$$
(3)$${\mathrm{Cu}}^{2+}+\mathrm{GSH}\to {\mathrm{Cu}}^{+}+\mathrm{GSSH}$$

### 2.1. Copper-Based Apoptosis

Apoptosis is a main form of cell death that exerts a pivotal role in the treatment of cancers. Excessive Cu can induce the toxic level of ROS that may aggravate the mitochondrial ROS, causing mitochondrial damage and cell apoptosis through a mitochondrial-mediated pathway. As it is known that the mechanism of apoptosis is complex and involves various signaling pathways, exploiting the primary target of main pathway to apoptosis is popular in treatment strategies. The abundance of research suggested that copper-based apoptosis in cancer was feasible. For example, Yang and colleagues demonstrated that copper sulfate (CuSO_4_) could induce apoptosis which was mediated by ROS in chicken hepatocytes via the mitochondrial pathway. Cu-induced ROS increased Bax (pro-apoptotic protein), while Bcl2 (anti-apoptotic protein) was decreased, resulting in altering mitochondrial membrane permeability by lowering mitochondrial membrane potential (MMP, an early stage of apoptosis) and releasing CytC that activated Caspase3 [14]. As immune cells exerts a vital role in clearing cancerous cells, Luo investigated the cytotoxicity of excessive Cu in RAW264.7 (macrophage cells). Their study suggested that CuSO_4_ induced both apoptosis and autophagy (a form of cell death) which were originally resulted from mitochondrial ROS. Interestingly, it was discovered that the inhibition of autophagy could increase apoptosis, suggesting that activated apoptosis protected RAW264.7 cells from Cu-induced apoptosis [15]. It was indicated that inhibiting autophagy to increase apoptosis might improve anticancer efficiency. Despite the mitochondrial pathway, apoptosis also occurs via the endoplasmic reticulum (ER) pathway. In most cases, ER-mediated apoptosis was induced by ER stress, which could subsequently activate PERK (transmembrane proteins of ER) or Caspase7 to cause apoptotic cell death. Liu et al. suggested that exposure to copper oxide nanoparticles (Nano-CuO) triggers OS by ROS, thus stimulating endoplasmic reticulum (ER)-stress and producing toxicity and apoptosis in male rat liver and BRL-3A cell. When ER stress occurred, ER transmembrane proteins including PERK, IRE1 and ATF6 were activated, which thereby enhanced the expression of CHOP (regulator of ER-mediated apoptosis protein) and CHOP-induced apoptosis by regulating apoptosis-related proteins (GADD34, ERO1 and Bcl family proteins) [3].

In addition, excessive Cu also induces apoptosis via death receptor signaling (transmembrane proteins that could be activated by outside factors and could lead to apoptotic cell death via various apoptosis pathways). For example, Guo et al. showed that disulfiram/copper complex (DSF/Cu) promoted apoptosis and prevented tumor development in human osteosarcoma via generation of ROS that activated the N-terminal kinase (JNK, a member of the MAPK family that plays an essential role in cell growth including ROS-mediated apoptosis) signaling pathway [16]. Recent research indicated that copper caused OS and apoptosis in hippocampus neurons by preventing cyclic adenosine monophosphate (cAMP) response element binding protein (CREB) phosphorylation. Interestingly, later research discovered that Cu-induced ROS could not only trigger apoptosis, but also activate the antioxidant protection signaling pathway nuclear factor erythroid 2-related factor 2 (Nrf2)/heme oxygenase 1 (HO-1)/NAD(P)H quinone dehydrogenase 1 (NQO1) [17]. In contrast to Yang’s study (the increase in SOD and CAT), Guo’s research showed that CuSO4 could decrease the activity of antioxidant enzymes including superoxide dismutase (SOD), catalase (CAT) and glutathione peroxidase (GPx). At the same time, CuSO_4_ induced apoptosis by increasing TUNEL-positive(apoptotic cell labeling) cells in the spleen and the generation of γH2AX (a DNA damage marker) [18]. Zha’s study demonstrated that DFS/Cu generated enhanced antitumor effect in myelodysplastic syndromes (MDS) by blocking cell cycles at the G0/G1 phase, up-regulating the level of P21 and P27 while down-regulating cyclin D1 and cyclin A2. Mechanistically, DFS/Cu exhibited anti-MDS effects via ER-mediated apoptosis and AKT inhibition [19]. Santos’ investigation of long-term Cu exposure indicated that the induction of OS inhibited CAT and GPx level, as well as dysregulation of apoptosis due to the elevation of caspase-3 [20]. These results show that copper complexes may trigger adaptive antioxidative signaling pathway when ROS-dependent apoptosis occurs. In comparison with commercial copper complexes, copper nanomaterials may be prone to induce lethal apoptosis by boosting ROS that makes cells unable to activate their self-repairing mechanism including antioxidative reaction. The illustration of Cu-based apoptosis is shown in Figure 1.

Excessive Cu+ induced OS and elevated ROS, which could activate the adaptive antioxidative signaling pathway (elevates the antioxidative gene expressed: Nrf2, HO-1, NQO1, SOD and CAT). The Cu- induced ROS-activated mitochondrial apoptotic pathway increases the concentration of intracellular CytC, and it induces apoptosis by activation of caspase3. In addition, excessive Cu could also increase ER stress, which could activate the PERK, IRE1 and ATF6 pathways, which would enhance the expression of CHOP and induce apoptosis by activating caspase3. Cu could also activate death receptor pathway by inhibition of AKT (regulating MMP by inhibiting BAD) and activation of JNK pathway (regulating MMP by inhibiting Bcl2 and activating BAX).

### 2.2. Copper-Based Autophagy

Autophagy is a kind of adaptive process that occurs before apoptosis [21]. Because autophagy and apoptosis are inhibitory in most cases, many cellular stress pathways activate autophagy early and subsequently induce cell apoptosis when the stress is overdone [22]. Guo et al. showed that CuSO_4_-induced ROS not only triggered protective autophagy via the AMPK-mTOR-dependent signal transduction pathway in the mouse-derived spermatogonia cell line, but it also promoted ferroptosis (another form of cell death that was related to iron metabolism), increasing the harmful effect of CuSO_4_. CuSO_4_ induced autophagy that downregulated mTOR expression and upregulated the expression of AMPK (AMP-activated protein kinase involving mitochondrial homeostasis) as well as ULK1 (membrane-targeted autophagy-activating kinase). Meanwhile, LC3 and p62 decreased (biomarkers of autophagy-related proteins) while autophagy-related proteins increased (ATG5-12, ATG7, ATG3 and Beclin1; involving autophagosome formation) [23]. Previously, Zhong et al. revealed that Cu complex also triggered protective autophagy in response to OS in Hela cells by activating the p38 MAPK pathway [24]. Researchers also found that activated protective autophagy had response to copper-induced apoptosis in ATP7B-deficient hepatocytes in Wilson disease [4]. Mechanically, as mTOR decreases, autophagy-related genes are activated by means of the translocated transcription factor EB of mTOR substrate to the nucleus, and the knockdown of autophagy-related genes: ATG7 and ATG13 can accelerate cell death.

Tsang’s study on autophagic process indicated that copper was a major regulator of ULK1/2, which was associated with autophagic flux, suggesting that Cu-chelation therapy in cancer could prevent cell proliferation by lowering autophagy [25]. However, the research performed by Zhang demonstrated that Nano-CuO was prevalently deposited in lysosomes of human umbilical vein endothelial cells (HUVECs), causing lysosomal dysfunction, blockage of autophagic flow, and cell death [26]. These results demonstrated that copper mostly induced cell death through autophagic pathway. In fact, autophagy induced by copper can either protect cells from death or contribute to cell death, depending on autophagic flux, which is associated with the concentration of copper. Thus, copper nanomaterials may potentially outperform conventional copper-based materials by causing more damage to autophagic flux by releasing large concentration of copper ion, thereby inducing apoptotic cell death. Recent study suggested that copper-based nano-alloy could reduce autophagy flux by effectively alkalizing lysosomes. In this study, the core–shell gold@copper selenide nanoparticles (Au@Cu_2 − x_Se NPs) can degrade H_2_O_2_ to generate O2 and increase pH (from 4.92 to 5.96) of the solution during reaction. It was discovered that Au@Cu_2 − x_Se NPs exerted no impact on lysosomes number and autophagosome-lysosome fusion, which could regulate metabolic autophagy process of autophagosomes with lysosomes; however, it changed lysosomal pH, explaining the inhibition of autophagy flux [27]. Another study validated the notion that inhibition of autophagy increased the antitumor effect on rat malignant glioma utilizing Cas III-ia (a copper compound with Cu^2+^ in center) [28]. As a result, copper nanomaterials combined with autophagy suppression will become a promising strategy for the treatment of copper-dependent malignancies. The illustration of Cu-based autophagy in shown in Figure 2.

Cu-induced OS and ROS could activate both AMPK and AKT signaling pathways, and activated AMPK downregulated mTOR pathway. However, activated AKT promoted mTOR signaling. After then, inhibited mTOR could block the deregulation of ULK1/2 that could enhance the autophagy process. Increased ROS also activated IRE1 pathway via ER stress, which could increase the expression of Beclin 1 by activating JNK signaling pathway via downregulated Bcl family that promoted autophagy. Elevated Cu-induced ROS also increased apoptosis that resulted in inhibition of autophagy.

### 2.3. Copper-Based Ferroptosis

Ferroptosis is a kind of iron-dependent lipid peroxidation-induced programmed cell death [29]. Copper-induced OS and ROS could lead to lipid peroxidation damage on cell membrane, which is an essential precondition for ferroptosis [30,31]. According to Li’s study, DFS/Cu exhibited an antitumor effect in nasopharyngeal cancer through ROS/MAPK and p53-mediated ferroptosis signal pathways. DSF/Cu elevated p53 expression level. Increased p53 activated SAT1, which induced lipid peroxidation via ALOX15, resulting in ferroptosis [32]. However, further study revealed that the effect of DSF/Cu could compensatorily increase Nrf2f (a transcription factor regulates cellular defense against OS), preventing cell death caused by DSF/Cu-induced ferroptosis [33]. These studies revealed that copper deprivation was an anticancer treatment strategy by increasing cancer cells ferroptosis. The accumulation of iron is one of the characteristics of ferroptosis, and copper can also influence the ferroptotic processes including iron homeostasis. A recent study explained the relationship between Cu and Fe in hepatocellular cancer. Cu accumulation inhibited HIF1α and upregulated ceruloplasmin (CP), which reduced Fe concentration and in turn inhibited lipid peroxidation and ferroptosis [34]. In addition, there was also crosstalk between autophagy and ferroptosis. According to Guo’s study, autophagy can aggravate CuSO_4_-induced cell death by promoting ferroptosis both in vivo and in vitro. Therefore, promoting autophagy and ferroptosis might enhance anticancer efficiency. In addition, recent study also constructed copper-based dual catalytic approach by combining artemisinin (ART) and copper peroxide nanodots to enhance autophagy and ferroptosis that produced highly cancer toxic reaction. Different from copper deprivation-induced ferroptosis, ART acted as an autophagy inducer and increased the intracellular iron concentration through degradation of ferritin, which could thus promote ferroptosis [35]. Collectively, the deprivation of copper can induce ferroptosis in cancer cells and activate the adaptive antioxidation pathway, causing resistance to treatment. Applying copper-based nanomaterial to boost toxic oxidative reactions or optimizing the treating environment to benefit the ferroptotic cell death might be conducive to resolving the dilemma of copper-induced therapy resistance. The illustration of Cu-based ferroptosis is displayed in Figure 3.

Cu-induced ROS upregulates p53 expression, which reduces expression of SLC7A11 (core subunit of system Xc^−^). Decreased SLC7A11 inhibits system Xc^−^ that cause inactivation of GPX4 and increase in lipid ROS. In addition, elevated p53 also could upregulate SAT1 expression, enhancing metabolism of polyunsaturated fatty acids (PUAFs) by increasing ALOX15. Cu accumulation upregulates CP to lower Fe concentration and inhibit lipid OS and ferroptosis.

### 2.4. Copper-Based Pyroptosis

Pyroptosis is a novel programed cell death (PCD) characterized by caspase-1 dependence, which can be initiated by various stress such as ROS via NLRP3, ILs and GSDMD, and can influence oncogenic process including proliferation, invasion, and metastasis [36,37,38,39] Several previous reports have revealed that copper can activate NLRP3 inflammasome. Deigendsch’s findings demonstrate that NLRP3 inflammasome activation in non-cancer cells requires intracellular copper [40]. In a recent study, Dong et al. showed that CuCl_2_ contributed to elevating NLRP3 inflammasome in primary microglia, as well as the level of CaspaseC-1 [41]. Mechanically, Zhou’s study indicated that CuSO_4_ could significantly promote pyroptosis via NLRP3/caspase-1/GSDMD axis. CuSO_4_ induced ROS-activated early survival signal of NF-κB pathway, which resulted in the downstream activation of NLRP3, IL-Iβ and IL-18, subsequently inducing pyroptosis. In addition, accumulated Cu also caused mitochondrial autophagy disorder by the downregulation of LC3 and p62, which could trigger pyroptosis via NLRP3/caspase-1/GSDMD axis [42]. ER stress controls cells fate through unfolded protein response (UPR), which is a crucial factor in cancer growth and invasion [43,44]. Based on Tardito’s study, copper overload generates ER stress and inhibits caspase-3 (noncanonical pathway of pyroptosis), whereas IRE1α-XBP1 pathway may be responsible for ER stress activation [45,46]. Cu-induced ER stress resulted in IRE1α and PERK activation, which could generate overexpression of CHOP. As a result, this caused cell apoptosis by activating Caspase3, which increased the downstream expression of GSDME (pyroptosis effector protein) and punched holes in the cell membrane, causing cell swelling and rapture, releasing inflammatory factors as well as resulting in pyroptosis. These results underlined the role of copper in pyroptosis, especially through caspase-1-dependent canonical pathway. Notably, ER stress is associated with tumor cell death mechanisms including ferroptosis, autophagy, apoptosis and pyroptosis [44]. The use of copper-based nanomaterials to produce excessive ROS could provide a prospective strategy for enhancing cancer cell injury by activating ER stress, enhancing tumor cells to be killed and overcoming therapeutic resistance. The illustration of Cu-based pyroptosis is shown in Figure 4.

Cu-induced ROS upregulates NLRP3 and inflammatory factors (IL-Iβ and IL-18), which activates Caspase1-dependent pyroptosis via activation of GSDMD. Cu-induced ROS also activates NF-kB pathway and mitochondrial autophagy (decreases LC3 and p62) that induces pyroptosis via NLRP3/caspase1/GSDMD axis. In the meanwhile, Cu enhanced ER stress activates caspase3-dependent pyroptosis.

### 2.5. Copper-Based Paraptosis

Paraptosis is a kind of PCD characterized by mitochondria an/or ER dilation, and cytoplasmic vacuolization via caspase-independent pathway that is different from apoptosis [47]. Paraptosis could be induced by ER stress as a result of DNA damage signal pathway [48]. As apoptosis resistance is intimately associated with tumorigenesis, paraptosis, as a substantial way following failure of apoptosis, could contribute to tumor cell death [49,50,51] Several studies have indicated that copper-based complexes can induce paraptosis in human cancer cells by inhibiting ubiquitin-proteasome system, involving ER stress and UPR activation. In an early investigation on cisplatin-resistant cancer cells, Marzono’s group found that cells treated with a copper-based complex did not show caspase-3 activation (classic executor enzyme of apoptosis), indicating a potential relation to trigger paraptosis [52]. Afterwards, they suggested that lysosome damage was an early cellular process of copper-based complex to overcome cisplatin-resistant cancer cells, whereas phosphine copper could suppress the growth of cancer cells via G2/M cell cycle and paraptosis, probably by boosting ROS [53]. Furthermore, the copper complex induced paraptosis in a wide panel of human cancer cell lines by upregulating UPR related genes, resulting in eIF2α phosphorylation, an increase in XBP1 mRNA, as well as overexpression of ATF4, CHOP, BIP and GADD34 [54]. The copper-based complex also exhibited tumor cells selectivity VS normal cells, and inhibited 26S proteasome activity, which could induce polyubiquitination and suppression of the ubiquitin–proteasome pathway, leading to ER stress [55,56]. Chen’s findings also supported the ubiquitin–proteasome-associated paraptosis, which depends on ATF4-based ER stress but is unrelated to ROS generation [57]. In a study of therapeutic resistance cancer cells, Chen et al. developed a DSF/copper complex copper diethyldithiocarbamate (Cu(DDC)_2_), inducing cell death in drug-resistant prostate cancer cells through paraptosis [58]. In general, copper-based complexes can cause cancer paraptosis by suppressing UPR and activating ER stress, representing a potential approach to overcoming cancer therapy resistance. However, there were few data concerning mechanisms of paraptosis. Therefore, more studies are required to identify key regulators related to Cu-based paraptosis.

### 2.6. Copper-Based Cuproptosis

Cuproptosis is a form of copper-dependent cell death. Recently, Tsvetkov et al. showed that intracellular Cu-induced a novel form of cell death by the aggregation of lipoylated mitochondrial enzymes and a loss of Fe–S proteins. To be specific, this process could not be inhibited by other known cell death pathway (necroptosis, ferroptosis and apoptosis). Based on this study, some researchers implied copper-based nanomaterials to proceed anticancer therapy in the form of cuproptosis. Lu et al. reported that copper-organic complex copper(II) bis(diethyldithiocarbamate) (CuET) decreased the expression of FDX1 (upstream regulator of mitochondrial protein lipoylation), suggesting that CuET triggers cuproptosis in A549 lung cancer cells [59]. Xu and colleagues investigated copper nanoplatform with glucose oxidase, GOx@[Cu(tz)], and induced cuproptosis in bladder cancer by oligomerization of lipoylated DLAT (a regulator of carbon that enters the TCA cycle that promotes cuproptosis) [60]. Jia et al. designed a brain-targeted nanoplatform with human H-ferritin (HFn), regorafenib and Cu-HFn-Cu-REGO NPs—thus increasing intracellular Cu concentration by elevating ATP7A (Cu exporter on cell membrane), while CTR1 (Cu importer on cell membrane) was lowered. Meanwhile, the level of mitochondrial ATP7B was downregulated, which further aggravated the accumulation of Cu, and aggregated cuproptosis by upregulating lipoylated DLAT and FDX1 [61]. The illustration of copper-based cuproptosis is shown in Figure 5.

Accumulated intracellular Cu^2+^ is combined with lipoylated DLAT that induces oligomerization of lipoylated DLAT, which promotes cell death by cuproptosis. FDX1 reduces Cu^2+^ to Cu^+^ that inhibits biosynthesis of Fe-S protein and induces cell death.

## 3. Multifunctional Applications Based on Copper-Based Cell Death

### 3.1. Copper-Complexes Chemotherapy

The widespread utilization of cisplatin successfully fights against cancer in clinical treatments. However, the anticancer activity was still limited by dose-limiting effects and inherited or acquired cancer resistance [62]. To improve this situation, researchers developed alternative strategies based on various metals. In this field, copper complex exhibited excellent anticancer effects similar to platinum-based complexes. Copper complexes with ligands, including 8-hydroxyquinoline, a copper chelator inhibiting proteasome activity, caused proliferation suppression and apoptosis in breast and prostate cancer cells [63,64]. Dithiocarbamate (DTCs) is another ligand with proteasome inhibition activity. Diethyldithiocarbamate (L^14a^) is the main product of Cu and DSF, and it also exhibits anticancer effects. Dou et al. proved that [Cu(L^14b^)_2_]14b could induce apoptosis in prostate and breast cancer cells by activating calpain and the Caspase 3-dependent pathway [65]. Chen et al. discovered that it activated JNK, NF-κB and Ap-1 in apoptosis [66]. In addition, Cu-MOFs including HKUST-1, Cu-2MI MOF and Sm-TCPP nanosheets have been explored in L^14a^-based chemotherapy. Gao et al. employed TCPP (tetrakis(4-carboxyphenly)porphyrin) and rare-earth element Sm, loaded Cu and hyaluronic acid (HA), Sm-TCPP(Cu)@HA, exhibiting high stability as well as Cu carry rates that intensified the anticancer activity of DSF [67]. Hou et al. produced HA modified Cu-MOF (ligand:2-MI). At the tumor site, it could be internalized through CD44 receptor-mediated endocytosis and decomposed under low pH conditions, thus releasing L^14a^ killed 4T1 breast cancer cells by binding NPL4 [68]. The above-mentioned studies suggested that Cu-MOFs might be a robust nanoplatform for enhancing chemotherapy activity of Cu-organic compounds.

### 3.2. Chemodynamic Therapy

When triggered by copper NPs-loaded antitumor-drug agents, chemodynamic therapy (CDT) relies on the in situ Fenton reaction, which is activated by endogenous media including GSH and H_2_O_2_, requiring no external energy input in the tumor region [69,70]. CDT circumvents the restrictions posed by the limited penetration of light through tissues. Recently, copper-based substrates can be categorized by endogenous and/or self-sufficient H_2_O_2_ due to their high efficiency in weakly acidic tumor microenvironments (TME). For instance, Liu et al. constructed endogenous H_2_O_2_-generated GSH activating nanoplatform (Ag@HKU-HA). This nanomaterial could catalyze endogenous H_2_O_2_ to toxic ·OH, and release Ag at the tumor site via degradation of external MOFs triggered by Cu^2+^-reduced glutathione, causing increased apoptotic cell death in vivo and in vitro [71]. Ma et al. demonstrated the utilization of self-assembled nanoparticles (Cu-Cys NPs) for in situ glutathione activation and H_2_O_2_ reinforcement in drug-resistant breast cancer. Following endocytosis, Cu-Cys NPs could interact with local GSH as an early response, depleting GSH. Then, Cu^+^ efficiently reacted with local H_2_O_2_ to produce toxic ·OH, which is responsible for cancer cell apoptosis. Notably, this reaction exhibited a high reaction rate under strongly acidic conditions (pH = 5), aiming to ensure the high efficiency of Cu-Cys NP-mediated chemodynamic therapy (Figure 6c) [72].

In addition, several studies proved that copper NPs enhance antitumor activity by increasing cancer cell apoptosis. Cu_6_NC was the cooper nanocluster that could gradually fracture and generate ROS when exposed to acidity, showing high cytotoxicity in cancer cells in a concentration-dependent manner but more moderated cytotoxicity in normal cells [73]. Shen designed three copper-based complexes based on quinoline derivatives, which were beneficial for promoting ER stress and caspase-dependent apoptosis in ovarian cancer cells [74]. Zheng et al. proposed that PEGylated CuMoO_x_-coated and zinc protoporphyrin IX (ZP)-loaded Cu (CCMZ) nanoparticles can suppress HO-1 (an important antioxidant NRF2 downstream molecular to eliminate ROS) activity and deplete GSH, resulting in apoptosis and necrosis [75]. Liu et al. synthesized TME-responsive calcium and copper peroxides nanocomposite CaO_2_-CuO_2_@HA NC with increased permeability and retention effect (EPR), where the synergistic effect of the Fenton reaction by Cu^2+^ and mitochondria dysfunction by Ca^2+^ could significantly increase the apoptotic cell death. CaO_2_-CuO_2_@HA NC induced mitochondria dysfunction by Ca^2+^ overloading caused by a much lower MMP (Figure 6b) [76]. Zhao et al. reported that a complex of gold nanozymes and DOX loaded with copper ion-doped ZIF-8 shell (ACD) could be degraded in the acidic TME. After degradation, the DOX and Au NCs were released, and the acidic pH with overexpressed GSH could activate fluorescence imaging. The enhanced cancer CDT was reacted by the Fenton reaction, in which Cu^2+^ depleted GSH through redox reaction and converted internal H_2_O_2_ to ·OH, thereby strengthening the effect of DOX by inhibiting drug flux and boosting drug sensitivity. The synergistic CDT caused apoptosis of cancer cells (Figure 6a) [77]. The research conducted by Shen revealed that new copper-based complex Cu(L4)_2_ and Cu(L10)_2_ could reduce catalase activity, thereby limiting H_2_O_2_ transfer to H_2_O due to enhanced CDT. At the same time, they could cause mitochondrial dysfunctions and ER stress, thereby impairing autophagic flux and increasing apoptosis in cancer cells [78].

**Figure 6 molecules-28-02303-f006:**
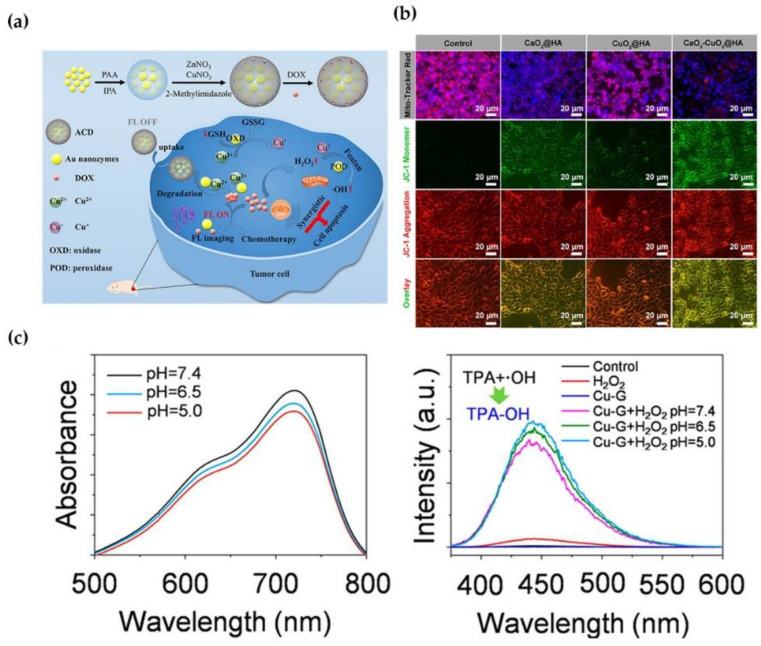
(**a**) Copper ions enhance CDT by depletion of GSH. (**b**) Cell apoptosis ratio of Cu^2+^ compared to Ca^2+^ in 4T1 cells. (**c**) Copper NP complex reacts quickly in slightly acidic TME. Reprinted with permission from Refs. (**a**) [77] 2021 American Chemical Society, (**b**) [76] 2022 American Chemical Society] and (**c**) [72] 2019 American Chemical Society.

Considering that the constant consumption of endogenous H_2_O_2_ in TME would diminish the efficiency of CDT, the involvement of extraneous enzyme could facilitate the insufficient H_2_O_2_, thus providing effective antitumor treatment. He et al. packed lactate oxidase (LOX) using cationic polyethyleneimine (PEI) on the basis of copper ions (PLNP^Cu^). The shell of PLNP^Cu^ was shed in an acid situation where lactate was be able to gather around. The LOX facilitated lactate to produce H_2_O_2_ that was continuously transformed into ·OH. Lactate consumption could produce H_2_O_2_, which was then transformed into antitumor ROS catalyzed by copper ions, causing immunological cell death and enhancing apoptosis. In this way, lactate depletion reversed the suppression of lactate-induced immune cell (Figure 7a) [79]. In addition, Li’s group also demonstrated that copper nanodots with DOX self-produced H_2_O_2_ in weakly acidic TME and generated ·OH through the Fenton-like reaction, as well as singlet oxygen (^1^O_2_) and superoxide anions (·O^2-^), thus accelerating cell apoptosis in breast cancer cells [80]. Fu et al. applied poly (ethylene glycol)-modified glucose oxidase (GO_x_) as an enzyme catalyst to effectively catalyze intracellular glucose and supply H_2_O_2_ for subsequent Fenton reaction. Meanwhile, released Cu^2+^ and endogenous GSH induced GSH depletion and Cu^2+^ reduction, which induced H_2_O_2_ to generate ·OH through a Cu^+^-mediated Fenton reaction, resulting in the improved CDT efficiency [81]. In addition, Meng’s results revealed a new paradigm that exerted CDT via the reduction in O_2_ consumption. Dox@Cu-Met NPs were DOX-loaded copper-metformin (Met) nanoscale coordination polymers, of which Met could inhibit O_2_ consumption to relieve TME hypoxia by suppressing mitochondrial respiration, thus elevating the concentration of H_2_O_2_ and improving DOX efficiency through the DOX-activated cascade reaction of NADPH-NOXs-SOD. In vivo bioactivity showed that Dox@Cu-Met NP-treated group had the lightest tumor tissue with relative tumor volume, exerting significant tumor inhibition effect (Figure 7b) [82].

Furthermore, Cu complexes can potentially be combined with other therapies to boost CDT efficiency. Liu et al. developed bismuth-based Cu^2+^-doped BiOCl nanotherapeutic platform BiOCl/Cu^2+^-H_2_O_2_@PVP (BCHN), which could self-supply and consume GSH to degrade the composite in weakly acidic TME, thereby generating ·OH through the Fenton reaction to achieve OS in CDT, and can also be combined with X-ray to realize bismuth-based sensitization radiotherapy. In vitro assay demonstrated that BCHN significantly reduced the colony-formation rate and cancer cells apoptosis, indicating that BCHN acted as a radiosensitizer. In slightly acidic TME, BCHN may release self-supply H_2_O_2_ to generate BCN (BiOCl/Cu@PVP), which is subsequently biodegraded by overexpressing GSH in tumor cells. The continuing production of ·OH and the consumption of GSH in tumor cells were boosted by modulating OS level. Furthermore, generated O_2_ could also relieve cancer hypoxia. (Figure 7c) [83]. Based on copper peroxide (CP), Liu et al. presented another CDT-boosting compound. CP was synthesized by converting coordinated H_2_O_2_ to Cu^2+^, which dissociated into Fenton catalytic Cu^2+^ and H_2_O_2_ in acidic environment of endo/lysosomes and released Cu^2+^ and H_2_O_2_. Through lipid peroxidation, the decomposed ·OH disrupted lysosomal membrane permeabilization, causing apoptosis with lysosome-associated pathway. In addition, CP nanodots also presented favorable enhanced EPR effect, significant tumor accumulation, and pH-dependent ·OH generation, which could effectively suppress tumor growth with minimal side effects in vivo.

### 3.3. Phototherapy

Phototherapy primarily involves photothermal therapy (PTT) and photodynamic therapy (PDT) [84]. PDT is a noninvasive therapy that adopts photosensitizers (PS) for generating cytotoxic ROS in premalignant and neoplastic diseases, while PTT aims to produce hyperthermia-induced cell death using photoabsorbing substances [85].

#### 3.3.1. Photodynamic Therapy

PDT primarily employs light, molecular oxygen (O_2_) and PS to generate highly cytotoxic ROS, which can induce cell death through type-I and/or type-II photochemical reactions [86,87]. In the type-I reaction, the excited triplet state of PS directly interacts with local biomolecules in cancer, generating radical cations or anions and subsequently reacting with O_2_ to produce cytotoxic ROS (O_2_^−^, ·OH, H_2_O_2_, etc.). In addition, for type-II reaction, excited triplet state of PS directly sensitizes O_2_ and generates ^1^O_2_ inside cancer cells [88]. Even though PDT has advantages in the treatment of certain superficial tumors including skin cancer and oral cancer, it remains challenging to address the limitations of high tumor selectivity and ROS production efficiency, as well as the lack of monitoring of treatment response. Over the past few decades, copper-based nanomaterials have provided an alternative approach to refining the performance of PDT. Wang’s team developed copper-doped carbon dots (Cu-CDs) with high fluorescence quantum yield of 24.4%, minimum cytotoxicity with high quantum yield of ^1^O_2_, as well as significantly inhibited 3D multicellular spheroids growth [89]. Guo et al. developed high-yield ^1^O_2_ with replace magnesium(ii) in chlorophyll that boosted electronic intersystem cross rate by designing special tumor-selective NPs, Cu(ii)Chl-HA NPs. Owing to the HA component, Cu(ii)Chl-HA NPs revealed a special receptor-mediated target performance in CD_44_-overexpressed cancer cells [90]. In addition, Ocakoglu et al. also applied Cu-chlorophyll-based photosensitizer (Cu-PH-A) exhibiting great selectivity in ovarian cancer cells with strong PDT effectiveness and imaging function when labeled with I^131^ [91]. In a multi metal-based corrole complexes, copper (III) corrole displayed hypertoxicity to cancer cells compared to iron corrole whether in dark or illumination, which could thus achieve greater biosecurity and a low-demand working environment [92]. Cu(^R^QYMP)(dppn), a copper-based *N,N,O*-tridentate Schiff with benzo[i]dipyrido[3,2-a;2′,3′-c]phenazine (dppn) moiety, also exhibited DNA cleavage and cell viability through caspase-3-based apoptosis via PDT-dependent ROS generation [93].

In addition, to overcome the limitation of inadequate ^1^O_2_ quantum yield and poor cancer selectivity, researchers developed a copper-based complex by combining type-I and type-II PDT with gene silencing. Liu et al. constructed an ultrathin 2D Copper (I) 1,2,4-Triazolate Coordination Polymer Nanosheet (Ce6-DNAzyme/[Cu(tz)]) based on a GSH-responsive and -photosensitive nanosheet-based ultrathin 2D nanocarrier of a nonporous coordination polymer. Ce6-DNAzyme/[Cu(tz)] could be discomposed by excessed GSH, causing the tumor-targeted DNAzyme release for catalytic cleavage of targeting mRNA (EGR-1). Notably, Ce6 group on these nanosheets generated ^1^O_2_ via type-II PDT under 660 nm laser, while type-I response was triggered by 808 nm laser irradiation, thereby producing effective antitumor activity in hypoxia tumor microenvironment (Figure 8a) [94]. Recently, Xu’s team has developed a new glucose oxidase (GO_x_)-engineered copper NPs, denoted as GO_x_@[Cu(tz)], that activated cuproptosis-based synergistic PDT in bladder cancer. GO_x_ could be initiated by GSH stimulation that proceeded efficient glucose depletion, whereas the depletion of glucose and GSH sensitized cancer cells to GO_x_@[Cu(tz)]-associated cuproptosis via increased intracellular H_2_O_2_ [60]. Chen et al. introduced a new generation of PDT sensitizer based on copper-cysteamine (Cu-Cy) activated by X-rays. Through expression of proliferating cell nuclear antigen (PCNA) and E-cadherin, Cu-Cy NPs inhibited cancer cell proliferation and migration in a dose-dependent manner, with no obvious toxicities in vivo [95]. Moreover, He et al. developed a 5-ALA PDT treatment by employing copper-doped calcium phosphate NPs (CCPCA NPs) that alleviated cancer hypoxia and supplied continuous catalase-induced oxygen, which could enhance PpIX concentration and extend METP of PpIX by shutting down PpIX outflow (by decreasing HIF1-α and FECH) and upregulating PpIX generation (providing 5-ALA and increasing ALAS) (Figure 8b,c) [96].

#### 3.3.2. Photothermal Therapy

Photothermal therapy (PTT) refers to a kind of antitumor treatment that relies on near-infrared (NIR) laser to achieve hyperthermic ablation in response to cancer irradiation. Mechanically, it was proceeded in a photothermal agent that absorbs light energy and also transforms it to heat energy exposed to the irradiated region, killing cancer cells by exceeding the cytotoxic threshold (42.5 °C), which exhibits high tumor selectivity and is minimally invasive without systemic effects [97]. Li’s team developed a nuclear-targeted PTT based on copper sulfide NPs (CuS NPs) in order to overcome the clinical challenge of cancer recurrence caused by residual cancer cells after surgery or chemo/radiation therapy. CuS NPs had the ability to directly target cancer cells and then induce in nucleus by modification of RGD and TAT peptides, thus heating cancer cell to exhaustive apoptosis through 980 nm NIR irradiation [98]. Chen et al. established an NIR-activated CuS nanoplatform (CuS-RNP/DOX@PEI) that directly delivers Cas9 RNP and DOX for synergistic antitumor therapy. This nanoplatform could be absorbed in cancer cells via endocytic process and NIR-triggered CuS (41 °C) to break double-strand DNA, thereby inducing the accelerated release of Cas9 RNP and DOX. During this process, Cas9 targeting HSP90α reduced cancer heat tolerance, which could therefore enhance the mild PTT effect (43 °C) [99]. Moreover, copper-based PTT can enhance radiation therapy (RT). Zhou’s team developed a CuS-based nanoplatform coated by polyethylene glycol (PEG-[(64)Cu]CuS) in anaplastic thyroid carcinoma. Tumor growth could be delayed by PEG-[(64)Cu]CuS-mediated combined RT/PTT, which could dramatically prolong survival of tumor-bearing mice in comparison with RT alone [100]. Yan et al. developed a PTT synergized immunotherapy strategy based on CuS-RNP@PEI carrying modified Cas9 targeting PTPN2, which could not only trigger NIR-mediated PTT but also cause the accumulation of CD8 T cell as well as the upregulation of IFN-γ and TNF-α [101].

In addition, copper-based NPs could synergize PDT effect. Zhang et al. developed transferrin-labeled Cu@Gd_2_O_3_, which triggered PTT effect in 808 nm laser irradiation, increased mimic-peroxidase activity, and promoted intracellular ROS accumulation by PDT, thus significantly inhibiting cancer growth. Moreover, it also demonstrated enhanced MRI contrast that could achieve more pathological information [102]. Metal-organic frameworks (MOFs) have attracted considerable attention recently due to their high specific surface areas and tailorable porosity [103]. Weng et al. prepared a Cu-BTC metal-organic framework Cu@CPP-800 with the highest photothermal conversion efficiency (48.5%) in 808 nm laser irradiation, which was three times higher than commercial indocyanine green (15.1%) [104].

The above-mentioned studies indicated that combination therapy might enhance anticancer effect in comparison with monotherapy. The therapeutic outcomes of chemotherapy are not typically due to tumor heterogeneity and dose-limited anticancer effect. The combination with other treatment model could efficiently kill cancer cells by different mechanisms of action. The hyperthermia released by PTT can not only kill cancer cells but also help release thermos-responsive drugs, as well as enhance cellular uptake of anticancer drugs.

## 4. Conclusions and Outlook

To conclude, the form of copper-based cell death and the application of Cu-complexes are reviewed. Copper imbalance could influence various diseases especially cancer. Increased copper concentration directly or indirectly contributes to all stages of cancer progress. There are two approaches to managing anticancer therapy including (1) depleting copper to limit its bioavailability and (2) supplying copper to induce specific cytotoxicity.

In the anticancer applications, numerous studies revealed that copper complexes induce all types of cell death by influencing various death-related genes and proteins. In particular, through combination therapy, several proteins/pathways are used to inhibit tumor growth. However, severe cytotoxicity and biosafety became the limitation of copper-based complexes application. Although combination therapy enhanced the anticancer efficiency, it required additional resource and costs. Therefore, it would be more attractive if single therapy could simultaneously influence multi-signaling pathways to inhibit tumor growth. Considering the essential role of many cancers-related molecules that regulate microelements, we envision the emergence of cancer organelle-targeted drugs, particularly depleting the microelement cofactors rather than wildly depleting them to achieve the targeted treatment with biosafety in the living organisms. Therefore, integration of empirical screening approaches with novel research emerging from genome and proteosome studies as well as bioinformation technologies might provide the most efficient and precise way to determine other promising targets and gain molecular insights into the mechanism of copper complexes application. In addition, the application of copper complexes temporarily inhibited tumor growth. Nevertheless, during the intermission, regain of microelement or molecular is another key issue that needs to be addressed.

Multifunctional copper-based nanomaterials have made significant progress in developing antitumor therapy during the previous decade. Compared with conventional copper complex, copper-based nanoplatforms provided efficient ^1^O_2_ quantum yield and promising energy absorption in NIR, which could enable better cancer selectivity and lowered cytotoxicity, thus showing outstanding imaging and antitumor agent to diagnose and prevent cancer recurrence. The obtained results showed that copper was involved in multi-cell death pathways causing devastating injury to cancer cells, including mitochondrial dysfunction, indicating that copper-based nanomaterials theragnostics was a prospective antitumor approach when combined with drug delivery systems such as tumor-targeted drugs with genetic modification. Moreover, this review provides a comprehensive and detailed overview of the existing cell death mechanisms and application of copper nanomaterials, which will shed some lights on future studies regarding copper-based materials.

## Figures and Tables

**Figure 1 molecules-28-02303-f001:**
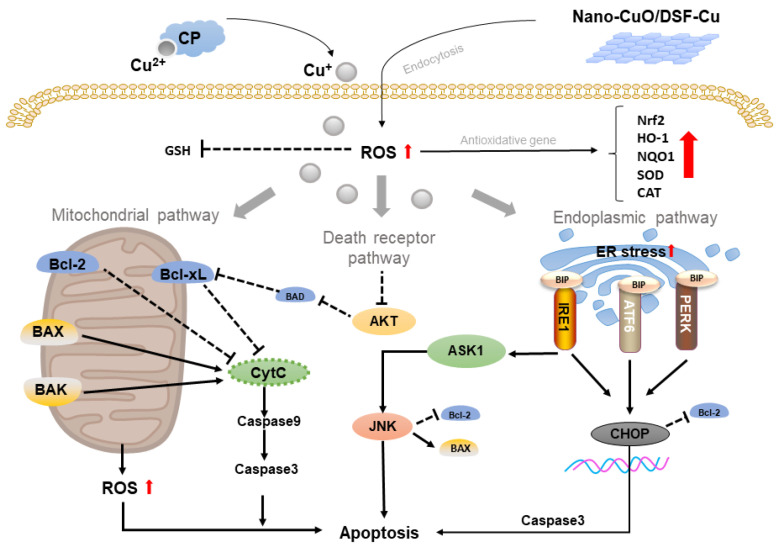
Schematic representation of copper-based apoptosis.

**Figure 2 molecules-28-02303-f002:**
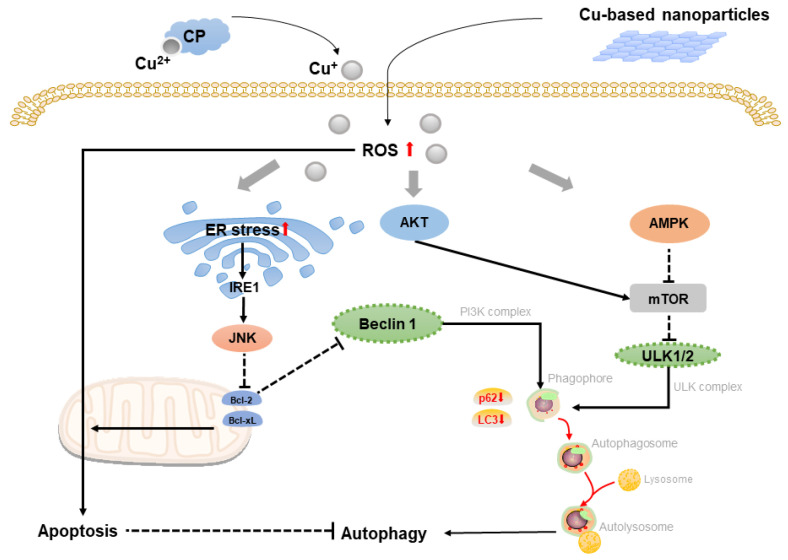
Schematic representation of copper-based autophagy.

**Figure 3 molecules-28-02303-f003:**
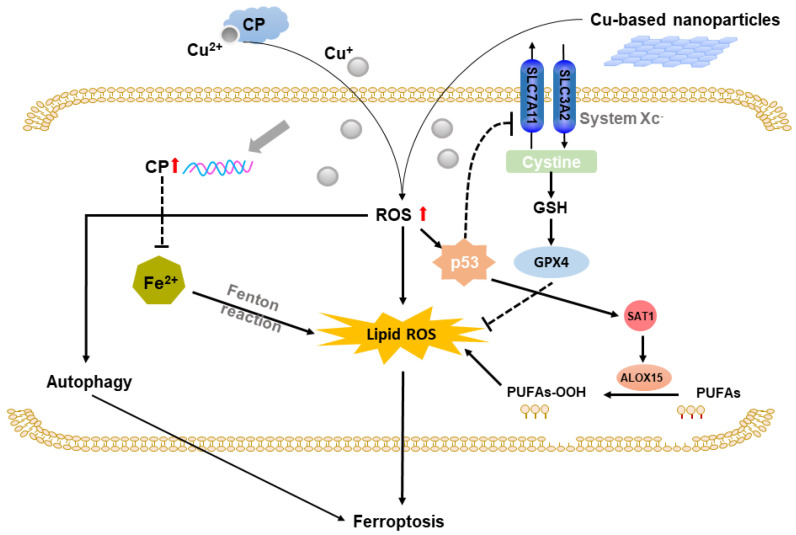
Schematic representation of copper-based ferroptosis.

**Figure 4 molecules-28-02303-f004:**
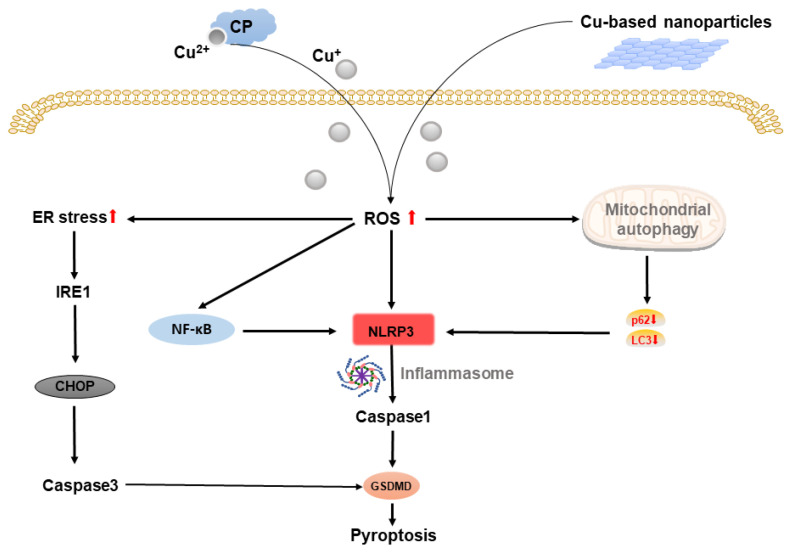
Schematic representation of copper-based pyroptosis.

**Figure 5 molecules-28-02303-f005:**
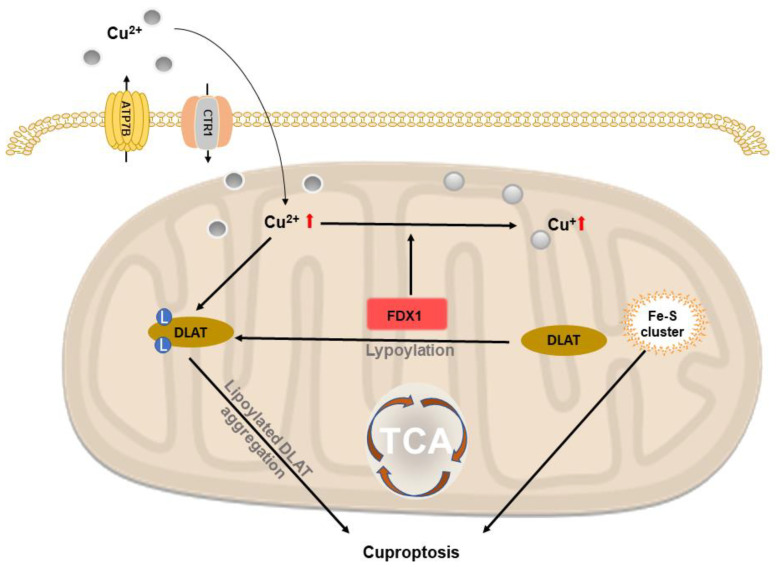
Schematic representation of copper-based cuproptosis.

**Figure 7 molecules-28-02303-f007:**
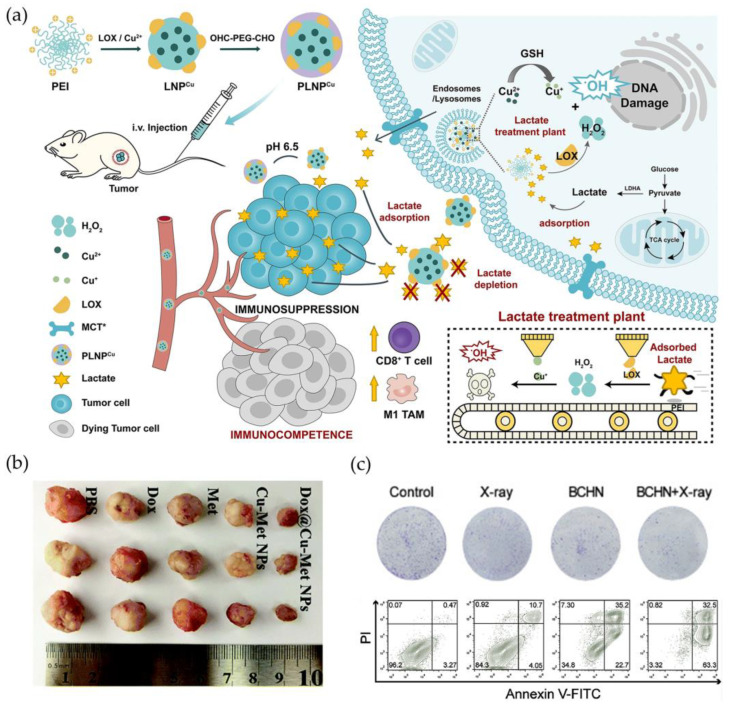
(**a**) Copper transforms lactate-produced H_2_O_2_ into antitumor ROS, mediating an immunogenic cell death. (**b**) Copper-based NPs complex significantly inhibits tumor with good biocompatibility. (**c**) Copper-based NPs complex reduced the colony-formation rate and cancer cells apoptosis, indicating that BCHN acted as a radiosensitizer. Reprinted with permission from Refs. (**a**) [79] 2021 Springer Nature, (**b**) [82] 2021 Royal Society of Chemistry and (**c**) [83] 2021 Elsevier.

**Figure 8 molecules-28-02303-f008:**
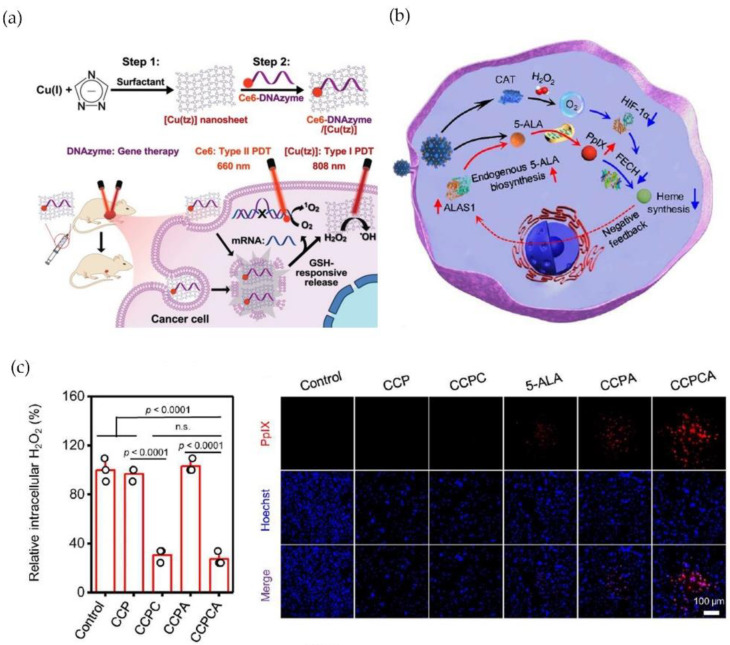
(**a**) Ce6-DNAzyme/[Cu(tz)] triggers both Type-I and Type-II PDT, as well as selective gene silencing by DNAzyme. (**b**) Illustration of heme negative feedback mechanism of CCPCA NPs. (**c**) CCPCA exhibited lower H_2_O_2_ content and RDPP expression (the study of intracellular O_2_), suggesting sufficient intracellular O_2_ generation ability (n.s. represents no signigicance). Reprinted with permission from Refs. (**a**) [94] 2021 John Wiley and Sons and (**b**,**c**) [96] 2022 Springer Nature.

## Data Availability

Not applicable.

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
