# Peer review of "Recent Advances in Cancer Therapeutic Copper-Based Nanomaterials for Antitumor Therapy"

_molecules, 2023, doi:10.3390/molecules28052303_

Round 1

Reviewer 1 Report

The manuscript proposes to review the literature regarding the use of copper-based materials as anticancer agents, describing the possible pathways for the tumor elimination to occur. 

In spite of the data presented by the authors being very relevant for the field, I am afraid the manuscript was poorly written in terms of English correction. There is a plethora of errors throughout the text which make it very confusing in some parts. Those errors are mostly lack of connective words, lack of subject-verb agreement, typos, etc.

The data extracted from the literature are simply piled up in the paragraphs, therefore I suggest that, in the new version of the manuscript, the authors should describe the literature data in more details, with more explanations about the cell death pathways and mediators, even dividing figure 1 into many figures, one for each cell death pathway described. The data from different sources ought to be connected in a more fluent way in the text.

Finally, I suggest reviewing the title, because the authors cite theragnostic nanomaterials, but the focus is on the therapeutic aspect, not the diagnostic one. 

Reviewer 2 Report

This topic on copper-based cancer therapy is of great interest but some important literatures and aspects are missing.

1. Elevated copper level in tumors dictate that either supplementing or depleting copper can exert anticancer activity. This aspect should be discussed in the introduction section. (Advanced Therapeutics 2019, 2 (5), 1800147. Nature Biotechnology 2021, 39 (3), 357-367. Adv. Colloid Interface Sci. 2022, 305, 102686.)

2. Like platinum-based chemotherapy, copper-organic complexes as anticancer agents could be used for cancer chemotherapy. This should be added as a section before section 3.1. Some relevant literatures such as Chem. Rev. 114 (1) (2014) 815–862. Biomaterials 2022, 281, 121335 should be cited.

3. Copper-based combinational therapy is more attractive than monotherapy. I would suggest the discussion of about it. Chemical Engineering Journal 2021, 409, 128222. ACS Nano 2021, 15(4), 6457-6470.

4. Cuproptosis induced by copper-based nanomaterials should be considered in section 2; some references are suggested; Journal of Materials Chemistry B 2022, 10 (33), 6296-6306. Advanced Materials 2022, 34 (43), 2204733.

Reviewer 3 Report

In this work, the authors have made a comprehensive and detailed 

overview of the antitumor bioactivity and the mechanis of tumor cell death caused by copper-based nanomaterials.After our careful evaluation, we recommended this work accepted in the Molecules after a minor revision.

 1.In the author's manuscript, when summarizing the part of "Multifunctional Applications Based on Copper-based Cell Death", the mechanism diagram and experimental diagram of the original author are quoted in detail. It is suggested to quote them in the overview of "Multifunctional Applications Based on Copper-based Cell Death" to ensure the consistency of writing!

 2.The author made a systematic and comprehensive overview of the research methods and progress of copper-based nanomaterials at present, but did not put forward the existing problems and solutions of copper-based nanomaterials in the field of tumor therapy. It was suggested that the author give an overview of the shortcomings and future development of copper-based nanomaterials.

Round 2

Reviewer 1 Report

The language and fluency of the manuscript has been improved, although there are still some improvements to be made. I believe the editors might be able to perform the language editing before the publication, though.

Overall, there are still some grammar issues, and it seems that there is no paragraph separation.

Regarding the figures, the authors were able to successfully create new figures, making it easier for the reader to understand the contents.

Author Response

Point 1: Overall, there are still some grammar issues, and it seems that there is no paragraph separation.

Response 1: Thank you for the detailed review. We have carefully and thoroughly proofread the manuscript to correct all the grammar and paragraph again.(marked in grean).We hope you will find this revised version satisfactory.

Reviewer 2 Report

Caption of Figure 5 should be changed into cuproptosis.

Author Response

Point 1: Caption of Figure 5 should be changed into cuproptosis.

Response 1: We were really sorry for our careless mistakes. Thank you for your reminder.